# Kinase Regulation of HOX Transcription Factors

**DOI:** 10.3390/cancers11040508

**Published:** 2019-04-10

**Authors:** Monika Primon, Keith D. Hunter, Hardev S. Pandha, Richard Morgan

**Affiliations:** 1Institute of Cancer Therapeutics, Faculty of Life Sciences, University of Bradford, Bradford BD7 1DP, UK; m.primon@bradford.ac.uk; 2Unit of Oral and Maxillofacial Pathology, School of Clinical Dentistry, University of Sheffield, Sheffield S10 2TN, UK; k.hunter@sheffield.ac.uk; 3Faculty of Health and Medical Sciences, University of Surrey, Guildford GU2 7XH, UK; h.pandha@surrey.ac.uk

**Keywords:** HOX, phosphorylation, embryonic patterning, cell cycle, cancer

## Abstract

The *HOX* genes are a group of homeodomain-containing transcription factors that play important regulatory roles in early development, including the establishment of cell and tissue identity. *HOX* expression is generally reduced in adult cells but is frequently re-established as an early event in tumour formation and supports an oncogenic phenotype. HOX transcription factors are also involved in cell cycle regulation and DNA repair, along with normal adult physiological process including stem cell renewal. There have been extensive studies on the mechanism by which HOX proteins regulate transcription, with particular emphasis on their interaction with cofactors such as Pre-B-cell Leukaemia Homeobox (PBX) and Myeloid Ecotropic Viral Integration Site 1 (MEIS). However, significantly less is known of how the activity of HOX proteins is regulated. There is growing evidence that phosphorylation may play an important role in this context, and in this review, we draw together a number of important studies published over the last 20 years, and discuss the relevance of phosphorylation in the regulation and function of HOX proteins in development, evolution, cell cycle regulation, and cancer.

## 1. Introduction

The *HOX* genes encode a family of homeodomain-containing transcription factors that play important roles in the early embryo, including the establishment of cell and tissue identity, and the regulation of cell proliferation, differentiation, and survival [1]. A characteristic feature of this gene family is their organisation within four clusters, each of which is located on a different chromosome. These clusters, A, B, C, and D, are used in the nomenclature of *HOX* genes, which are also numbered according to their relative position in the cluster, with, for example, HOXB1 being the most 3’ member of the B cluster [2]. Despite being originally characterised as developmental genes, the HOX transcription factors are known to have additional roles in the adult, including, for example, the proliferation of hematopoietic stem cells (HSCs) [3], and the maintenance of tissue identity during the menstrual cycle [4]. 

The highly conserved homeodomain of HOX proteins mediates their binding to DNA, although the strength and specificity of this interaction is greatly increased by the binding of co-factors such as Pre-B-cell Leukaemia Homeobox (PBX), which forms heterodimers with HOX proteins in groups 1-10, and Myeloid Ecotropic Viral Integration Site 1 Homolog (MEIS) proteins that dimerize with HOX proteins 9–13 [5]. These cofactors have a role in in the recruitment of RNA polymerase II or III, as well as transcriptional inhibitors such as histone deacetylase (HDAC), resulting in differential gene regulation depending on the sequence and context of the target site in the enhancer or promoter region [6,7].

Many of the *HOX* genes are also highly over-expressed in a range of cancers including melanoma [8], and head and neck [9], prostate [10], breast [11], ovarian [12], and pancreatic cancer [13]. In this context they usually have a pro-oncogenic role, supporting a malignant phenotype. The latter includes promoting proliferation and blocking apoptosis [14], the induction of angiogenesis [15], and facilitating metastasis [16], drug resistance [11,17,18], and radiation resistance [19]. The key roles that HOX proteins play in cancer make them potential therapeutic targets, although a high level of functional redundancy amongst HOX proteins presents a barrier to this approach. An alternative strategy is to inhibit the interaction between HOX and PBX, which is mediated by a conserved hexapeptide sequence in HOX proteins. This interaction can be inhibited using HXR9, a small peptide mimic of the hexapeptide sequence that causes apoptosis in a range of cancers [20]. 

Given the profound homeotic activity of *HOX* genes and their ability to influence processes in the adult, including DNA repair and cell cycle regulation, there is relatively little known of how HOX activity is itself regulated. Regulation occurs at the level of transcription in a process that is influenced by enhancer sharing as well as epigenetic changes and signalling through a range of pathways, most notably retinoic acid [1]. There have also been many studies of the role played by HOX co-factors such as PBX and MEIS [20]. However, relatively little is known of how HOX transcription factors are regulated at the post-translational level, despite HOX proteins containing multiple consensus sites for a range of kinases (Figure 1). There was a recent, excellent review on HOX post-translational modifications, which included phosphorylation [21]. Here, however, we focus on what is currently known about HOX protein phosphorylation with special emphasis on its functional consequences in development, cancer, and cell-cycle regulation.

## 2. Kinase Regulation of HOX Proteins in Development and Evolution

As discussed above, the *HOX* genes play a key role in the patterning of both the vertebrate and invertebrate body plan. In the latter, there is growing evidence that variations in HOX proteins might have helped drive evolutionary changes in the morphology and number of appendages. A particularly well-studied example is the change in the Ultrabithorax (UBX) protein that increased its ability to block transcription of Distal-less (Dll), a gene that also encodes a homeodomain-containing transcription factor. Dll is required for the development of appendages along with other embryonic structures, but the change in UBX function to a transcriptional repressor of Dll is considered to have driven a key change in the body plan of invertebrates, away from a uniform pattern of segments each bearing a limb (as typified by the centipedes), to a more complex body plan exhibited by most insects [22]. This change in UBX function may have been mediated by the loss of serine and threonine residues at casein kinase 2 (CK2) phosphorylation sites; the sites are present in invertebrates that have uniform Dll expression, but absent in those that lack Dll expression in UBX expressing cells (Figure 2). Furthermore, reintroducing CK2 phosphorylation sites in Drosophila UBX prevents it from repressing Dll, resulting in embryos that show signs of multiple limb development [23]. The phosphorylation of UBX may, therefore, play a key role in the repression of Dll expression and hence limb development in the abdomen. Subsequent studies have revealed CK2 phosphorylation to also regulate the transcriptional activity of other homeodomain-containing transcription factors including Engrailed-2 (EN2) [24] and Antennapedia (ANTP) [25]. CK2 phosphorylation of the EN2 homeodomain increases its DNA binding affinity 2–4-fold [24], whilst in the case of ANTP, it prevents binding to the extradenticle (EXD) cofactor and thus reduces its activity [25]. Changes to ANTP that prevent CK2 phosphorylation severely perturb thoracic and abdominal development, suggesting that it is no longer phenotypically suppressed by more posterior *HOX* genes. Conversely, acidic amino acid substitutions at the CK2 target site (mimicking a constitutively phosphorylated ANTP protein), greatly reduce its in vivo activity. Hence, as with UBX, CK2 phosphorylation plays a key role in modifying the ability of this homeoprotein to affect embryonic development [25]. 

CK2 consensus sites are also present in many vertebrate HOX proteins, indicating that these may also be regulated by CK2 as part of a developmental pathway (Figure 3). This is supported by the finding that the CK2 sites in HOXB7 are potential modifiers of its regulatory activity during the proliferation and differentiation of primary hematopoietic cells [26]. Mutations in the CK2 sites located in the first 14 amino acids of the C-terminal domain of HOXB7 enhanced the differentiation of the murine myelomonocytic cell line, 32D, which concurs with reports that CK2 can act as an oncogene when transfected into a number of normal mouse cell types (discussed in greater detail below) [27,28].

Phosphorylation also plays a key role in the regulation of another Drosophila *HOX* gene, sex combs reduced (SCR), which determines the identity of the labial and prothoracic segments. A study based on yeast two-hybrid screening revealed that the N-terminal arm within the homeodomain was a target of phosphorylation by cAMP-dependent protein kinase A (PKA) and dephosphorylation by protein phosphatase 2A, with the later activating and the former inactivating SCR function in vivo, through the modulation of DNA binding. Correspondingly, knockdown of the dPP2A,B’ gene encoding protein phosphatase 2A prevented salivary gland development, mimicking the SCR null phenotype [30].

An additional example of phosphorylation regulating HOX function is provided by the role of Abd-B in the induction of posterior spiracle organogenesis in Drosophila [31]. In the early stages of this process, Abd-B activates multiple downstream targets including the Janus Kinase/Signal Transducer and Activator of Transcription proteins (JAK/STAT) signalling pathway. However, at later stages, STAT activity feeds back directly into Abd-B, increasing its ability to activate transcription of downstream genes, whilst also acting to block the activity of other repressor proteins that inhibit the function of these target genes. Thus Abd-B/STAT signaling forms a cooperative loop to maintain a stable pattern of gene expression required for spiracle formation.

Phosphorylation is also an important regulator of the Caenorhabditis elegans HOX gene lin-39, a homologue of the ANTP gene of Drosophila and the HOX paralogues in vertebrates [32], which acts as a determinant of whether the epithelial Pn.p cells adopt a vulval precursor cell fate or fuse with the surrounding hypodermis (the F fate). In this context, the vulval fate is determined by lin-39 activity that, in turn, is enhanced through let-23 (receptor tyrosine kinase)/let-60 (Ras homologue) signalling, most likely leading to direct phosphorylation of lin-39 by MAPK [33]. 

## 3. Kinase Regulation of HOX Proteins during the Cell Cycle

In addition to a role in development and disease, there is growing evidence that HOX proteins play a role in regulating the cell cycle, and that they, in turn, can be regulated by cell-cycle dependent activities. One of the best-characterised examples of the former is the inhibition of HOXC10 transcriptional activity through binding of its homeodomain by the cell-cycle regulator Geminin [34]. Structural studies revealed that the homeodomain of HOXC10 binds to Geminin, an interaction that is mediated through the C-terminus of the latter in which the side chains of glutamates and aspartates generate an overall charge pattern similar to the DNA phosphate backbone. This interaction with Geminin is specific to HOXC10 as it is dependent on residues R43 and M54 in helix III, as well as a cluster of basic amino acids in the N terminus. The strength of the interaction between Geminin and HOXC10 can be increased through the phosphorylation of a C-terminal serine residue by CKII, resulting in enhanced inhibition of HOX transcriptional activity, and indicating an additional layer of regulation [34].

In addition to CK2, HOX proteins have also been shown to be substrates for cyclin-dependent kinases (CDKs) through phosphoepitope antibody recognition and electrophoretic mobility shift assays in Xenopus embryos. These screens revealed that HOXD1 is phosphorylated by the CDK cyclin B-Cdc2 during mitosis [35]. Another HOX protein that undergoes post-translational modification in a cell-cycle-dependent manner is HOXC10, which is targeted for degradation early in mitosis by the ubiquitin-dependent proteasome pathway [36]. Notably, amongst the abdominal-B related proteins, this property seems unique to HOXC10, as the levels of the paralogous HOXD10 and the related HOXC13 protein were, in contrast, found to be constant throughout the cell cycle. This HOXC10-specific activity appears to be dependent on two destruction box (D-box) motifs, as mutating these sequences results in the stabilisation of HOXC10, and the accumulation of cells in the metaphase. These findings indicated that HOXC10 is a prometaphase target of the anaphase-promoting complex (APC), which is supported by the co-immunoprecipitation of HOXC10 with the APC subunit CDC27, and its stabilisation in APC-depleted extracts. Hence HOXC10 has the potential to influence mitotic progression and may be a link between developmental regulation and cell cycle control [36].

Although there is currently no direct experimental evidence for a role of other kinases in regulating the cell-cycle dependent function of HOX proteins, multiple HOX family members have consensus recognition sites for the ATM serine/threonine kinase (Figure 1). ATM is best characterised for its role in the repair of double-stranded breaks in DNA, although it also interacts with proteins involved in the G1/S, intra-S and G2/M checkpoints [37]. Consensus recognition sites are also present for the functionally-related kinase ataxia telangiectasia and Rad3-related protein (ATR). ATR is activated in response to persistent single-stranded DNA found at stalled replication forks and causes cell cycle arrest, allowing repair to occur [38].

## 4. Kinase Regulation of HOX Proteins in Disease

Given the importance of the *HOX* genes in cancer, it is not surprising that HOX proteins are substrates of kinases known to have pro-oncogenic roles. These include the mammalian target of rapamycin (mTOR), a serine/threonine kinase that has a key role in cancer, promoting cellular activities that include protein synthesis, autophagy, survival, proliferation and growth. A proteomics-based study revealed that multiple HOX proteins (HOXA3, A5, A9, A11, B6, C4, and D11) can be phosphorylated by mTOR at a serine or threonine residue close to the N-terminus of each protein [39]. HOX proteins also contain consensus binding sites for another protein kinase with a key oncogenic role, PKA (Figure 1), which has been shown to have a role in the initiation and progression of multiple tumour types [40]. In a number of HOX proteins, there is a PKA consensus site either adjacent to or overlapping with the hexapeptide domain, indicating that PKA might also play a role in modulating the interactions between HOX proteins and their cofactors. Additionally, or alternatively, PKA might increase the transcriptional activation by HOX proteins through a similar process to that identified for the HOX co-factor MEIS1A [41], in which PKA phosphorylation increases its ability to bind the transducer of regulated CREB activity (TORC) protein. A role for PKA in regulating HOX activity is further supported by the observation that inhibition of PKA leads to reduced HOXD13 expression [42]. 

As discussed above, the disruption of HOX binding to PBX cofactors using a peptide antagonist of this interaction (HXR9) is cytotoxic in a range of solid malignancies, primarily through the induction of apoptosis [20]. The inhibition of HOX/PBX dimers also causes cell death in acute myeloid leukaemia (AML), although this seems to involve necroptosis (a regulated form of necrosis), rather than apoptosis [43]. The killing of AML cells by HXR9 is greatly enhanced by the inhibition of protein kinase C (PKC). Although the mechanism underlying this interaction has not yet been dissected experimentally, there are multiple PKC consensus sites in a large subset of HOX proteins, suggesting that PKC phosphorylation might directly affect HOX function through either DNA or cofactor binding, or stability [43]. The former is supported by the observation that HOXA9, identified as a key oncogene in AML, is phosphorylated at S204 and T205 by PKC and that this, in turn, increases the stability of the HOXA9/PBX/DNA complex [44]. 

The abundance of CKII sites in HOX proteins (Figure 3) indicates that this kinase also might have an important role in HOX protein regulation beyond early development, including cancer. CKII is a relative latecomer with respect to its potential as a therapeutic target, but it is known to be over-expressed in a wide range of both solid cancers and malignancies and supports tumour growth through regulating multiple cellular processes, including apoptosis and the cell cycle [45]. A study by Yaron et al. showed that CKII has a role in regulating HOXB7 function in the murine myelomonocytic cell line, 32D. Wild type HOXB7 blocks the G-CSF-induced differentiation of 32D cells, whilst HOXB7 with mutations that prevent PBX binding does not. Intriguingly, the mutation of 2 CKII sites in HOXB7 resulted in even more potent inhibition of cell differentiation compared to the wild-type protein, indicating that CKII acts as a negative regulator of this HOXB7 function [26]. 

During the development of both solid tumours and haematological malignancies, the HOX genes can form chimeric oncogenes as a result of chromosomal rearrangements at a relatively high frequency, although chimeric HOX proteins are more common in haematological malignancies [46]. This will clearly have profound effects on the post-translational regulation of HOX proteins as part of the protein will be missing, and the remaining portion will be fused to a different protein subject to different modifications. Furthermore, the chimeric protein will have a different tertiary structure that could hide or reveal new sites. A frequent fusion partner for HOX genes is NUP98, which encodes a nuclear membrane protein involved in the selective transport of RNA. Typically, NUP98/HOX fusions involve the C-terminal half of the HOX partner and the approximate N-terminal two-thirds of NUP98 [47,48]. Consequently, the kinase consensus sites described above for which experimental evidence exists (CKII and PKC) remain in the chimeric protein. In this context, it is noteworthy that both PKC [49] and CK2 [50] inhibitors have proved effective in AML, although it remains to be determined whether AML cells are expressing a HOX-containing chimeric protein and relatively more sensitive to these inhibitors.

## 5. Conclusions

There have been a number of important studies on the phosphorylation of HOX proteins that may have been overlooked due to their somewhat disparate nature and the fact that their publication was spread across several decades. Taken together though, the available evidence points to a significant role for phosphorylation in the regulation of HOX transcription factors in both development and cancer. There are also multiple, conserved kinase consensus sites in the HOX protein family which as yet have no known function, indicating a significant gap in our current knowledge of how these proteins regulate fundamental cellular processes. 

## Figures and Tables

**Figure 1 cancers-11-00508-f001:**
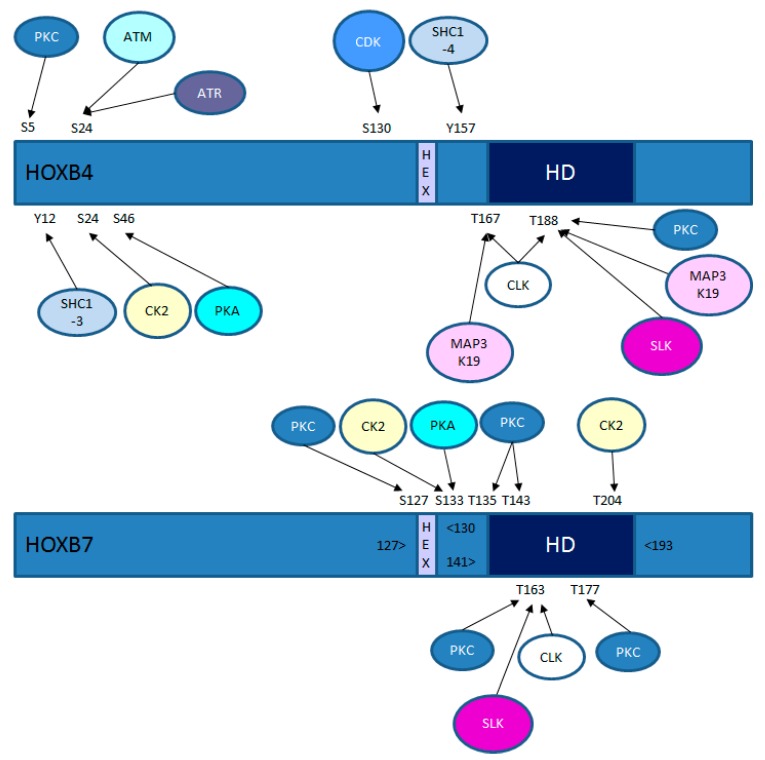
Consensus kinase sites in HOXB4 and HOXB7. The relative positions of the conserved hexapeptide domain (“HEX”) that mediates Pre-B-cell Leukaemia Homeobox (PBX) binding and the homeodomain (“HD”) that mediates DNA binding are shown. ATM, kinase mutated in ataxia telangiectasia; ATR, ataxia telangiectasia and Rad3-related protein; CK2, casein kinase 2; CDK, cyclin-dependent kinase; CLK, CDC2-like kinase; MAP3K19, mitogen-activated protein kinase kinase kinase 19 (also known as RCK and YSK); PKA, protein kinase A; PKC, protein kinase C; SLK, STE-20 like serine/threonine protein kinase.

**Figure 2 cancers-11-00508-f002:**
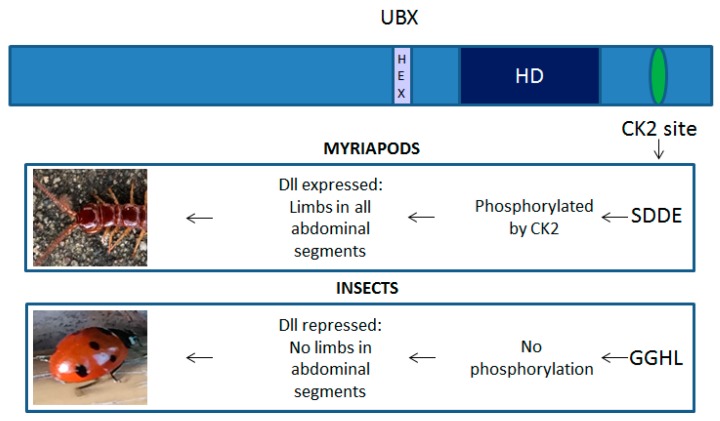
Casein kinase 2 (CK2) phosphorylation is a key regulator of Ultrabithorax (UBX) function. Phosphorylation of UBX by CK2 prevents it from repressing Ditsal-less (Dll) transcription in lower arthropods and consequently all embryonic segments posterior to the head give rise to limbs. This CK2 site is lost in insects and as a result UBX represses Dll expression, leading to the suppression of limb formation in the abdominal segments.

**Figure 3 cancers-11-00508-f003:**
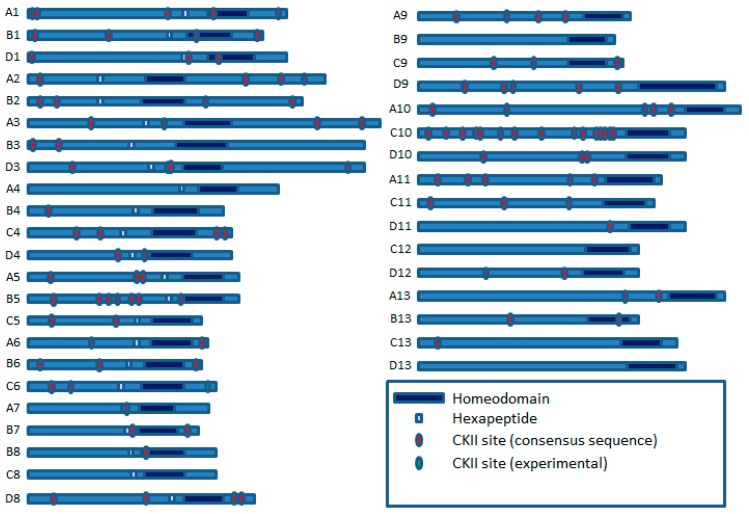
Potential Casein Kinase 2 (CK2) sites in human HOX proteins, based on the presence of the CK2 consensus sequence ((S/T)XX(D/E)) [29].

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
