# Peer review of "Kinase Regulation of HOX Transcription Factors"

_cancers, 2019, doi:10.3390/cancers11040508_

Round 1
Reviewer 1 Report
Primon et al. provide an overview of work on the question to what extent protein phosphorylation contributes to the regulation of HOX proteins. The article is organised into five sections. The introduction summarizes current knowledge about the HOX cluster’s genomic organisation, gene regulation and function in development and disease and, briefly, target motifs for various known protein kinases. This is followed by three sections meant to cover kinase regulation of HOX proteins during development, the mitotic cell cycle and disease. A brief conclusion underlines the paucity of data on HOX protein phosphorylation in normal and malignant cells.
Most of the introduction describes the genomic organisation of the HOX clusters, the roles of individual loci, how HOX proteins bind DNA and how some of these genes are de-regulated in cancers. These aspects have already been covered by a number of excellent recent reviews. It would perhaps be more informative to focus on examples of how well studied kinases, for which the HOX proteins shown in Figure 1 contain consensus motifs, influence the activity of DNA binding transcription factors.
I find section 4 on kinase regulation in disease confusing because it appears to describe a case of a HOX gene regulating a kinase, rather than the other way round. It is unclear how this pertains to the question, which kinases control HOX proteins and how these protein modification enzymes might exert their function.
The authors fail to mention a relevant recent study of the mammalian phosphoproteome that identified novel mTOR-dependent phosphosites in several HOX proteins (Kang et al., Proteomics 2017; see Supporting Information Table 4). Given this result, a timely review should explore the potential importance of mTOR signaling for HOX proteins.
Critically, a very recent review article on HOX proteins (Draime et al., Int J Dev Biol 2018) that comprehensively covers post-translational modifications, including protein phosphorylation, is not discussed. This is a serious omission.
Author Response
Many thanks for your helpful comments. We have shortened the introduction and removed the paragraph on the regulation of a kinase by a HOX protein, as you suggest. We have also discussed mTOR as a possible regulator of HOX function, and now refer to the review by Draime et al in the Introduction.
Reviewer 2 Report
Authors here reviewed the role of HOX family in development, cell cycle regulation and cancers. Especially, biological relevance of phosphorylation on HOX transcription factors are documented. Well written and summarized. I feel this review is quite informative in this field.
Several minor points point should be addressed before publication.
1) Literature review contents for section 3, kinases in regulating cell cycle is not sufficient (only 3 citations and no previous study on this topic). Since HOX gene has consensus site for DNA response proteins, perhaps the information of kinases in regulating DNA damage could be more comprehensive.
2) The title of each section is misleading. This review focuses on the role of kinases in regulating HOX genes/function. Therefore, the titles should be:
· Section 2: Kinase regulation of HOX gene in development and evolution
· Section 3: Kinase regulation of HOX gene in cell cycle
· Section 4: Kinase regulation of HOX gene in diseases
Furthermore, in section 4, the authors have mentioned about the role of HOXC10 in activation of CDK7 that is required for phosphorylation of RNA polymerase. This point could raise confusion to reader as the whole review is talking about how kinases affect the regulation HOX gene expression and function.
3) In introduction, authors mentioned HOX-overexpression in several cancers. It would be worthy to introduce not only over-expression but also fusion-mutation NUP98-HOX family in leukemia.
Rio-Machin A, et al.Leukemia. 2017 Sep;31(9):2000-2005.
Funasaka T, et al. Cell Cycle. 2011 May 1;10(9):1456-67.
4) There are some inconsistent font regarding HOXC10 in page 6, these must be revised.
Author Response
Many thanks for your helpful comments.
1. We agree that the section on cell cycle control is relatively short (although actually 5 citations, not 3), however it represents (as far as we are aware) all of what is known in this area. We still feel it is best included as a separate section though as the findings are important but do not fit very well into either of the other sections.
2. We have changed the subtitles as you suggest, and have deleted the section on the regulation of a kinase by a HOX protein.
3. We now include a discussion on the NUP98 gene fusions together with the suggested references.
4. The font inconsistencies have been addressed.
Reviewer 3 Report
The authors provide a good and well written review on regulation of HOX transcription factors activity by phosphorylation. Nevertheless, there is very little proven connection between this mechanism and cancer and obviously no application in clinical practice. In consequence, it would be more appropriate for this review to be published in a more generalist journal or in a journal on development.
Author Response
Thank you for your kind words. We have now extended the discussion around the role of kinase regulation of HOX proteins in cancer, making the review more relevant to this journal.
Round 2
Reviewer 1 Report
The revised manuscript takes the major issues into account.
Reviewer 3 Report
The authors took into account the suggestions for the revision of the manuscript, I have nothing more to add.